# The magic of top quarks

**Chris D. White**[1*] **and Martin J. White**[2†]

**1** Queen Mary University of London, London, UK
**2** University of Adelaide, Adelaide, Australia

★ christopher.white@qmul.ac.uk , † martin.white@adelaide.edu.au

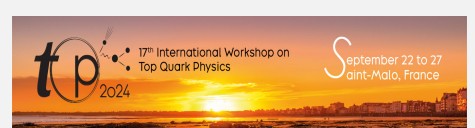

*The 17th International Workshop on
Top Quark Physics (TOP2024)
Saint-Malo, France, 22-27 September 2024*

## Abstract

In recent years, there has been increasing collaboration between the fields of quantum computing and high energy physics, including using LHC processes such as top (anti-)quark pair production to perform high energy tests of quantum entanglement. In this proceeding, I will review another interesting property from quantum computing ("magic"), that is needed to make quantum computers with genuine computational advantage over their classical counterparts. How to make and enhance magic in general quantum systems is an open question, such that new insights are always useful. To this end, I will show that the LHC naturally produces magic top quarks, providing a novel playground for further study in this area.

# 1    Introduction

In recent years, an increasing number of people have looked at using high energy colliders to test fundamental properties of quantum mechanics. Perhaps the most well-known quantum concept is that of *entanglement*, which clearly distinguishes quantum behaviour from classical, as encoded e.g. via Bell inequalities [1]. A particularly useful system for studying entanglement at the Large Hadron Collider is that of top (anti-)top pair production, examined in this context in e.g. refs. [2–7] (see also ref. [8], and ref. [9] for a critical appraisal of such measurements). Entanglement is, however, not the only special property of quantum states. Lots of other things are studied in either Quantum Computation or Information theory, for a variety of interesting reasons. Might these also be useful for high energy physics? In this proceeding, we examine one such property – *magic* – and argue that the LHC indeed offers an interesting situation for studying it. To introduce magic, let us first briefly review aspects of quantum computing.

# 2    A bit of quantum computing

In quantum computers, classical bits (with values $\{0, 1\}$) are replaced with *qubits*, namely normalised quantum states

$$|\psi\rangle = \alpha|0\rangle + \beta|1\rangle, \quad |\alpha|^2 + |\beta|^2 = 1, \tag{1}$$

where $|0\rangle$ and $|1\rangle$ are orthogonal basis states, and $\alpha$, $\beta$ complex coefficients. The canonical example of a single qubit system is a spin-1/2 particle, in which case $|0\rangle$ and $|1\rangle$ may represent the two linearly independent spin states. Multi-qubit systems can then be described using a basis comprised of tensor products of single-qubit states. Quantum computers take (multi-)qubits, and subject them to unitary transformations, where unitarity corresponds to conservation of probability in quantum theory. Each transformation acts on a given number of qubits, and is known as a *quantum gate*. These are the equivalent of *logic gates* in classical computing, and the quantum versions have fancy names such as *Hadamard*, *phase, CNOT* and *Pauli*. Precise details may be found e.g. in ref. [10], or ref. [11] in the context of this proceeding.

# 3    Could it be magic?

Quantum computers are expected to vastly outperform their classical counterparts, which is naïvely due to the two quantum properties of *superposition* and *entanglement*. However, it turns out that this is not quite true, and to see why, we need the concept of a *stabiliser state*. These are multiqubit states that give a simple spectrum for a restricted set of operators known as *Pauli strings*:

$$\mathcal{P}_n = P_1 \otimes P_2 \otimes \ldots \otimes P_N, \quad P_a \in \{\sigma_1^{(a)}, \sigma_2^{(a)}, \sigma_3^{(a)}, I^{(a)}\}. \tag{2}$$

In words: a Pauli string acting on $n$ qubits operates on the $a^{\text{th}}$ qubit with a Pauli matrix, or an identity matrix. There are then $4^n$ such strings for $n$ qubits. Stabiliser states are such that the set of expectation values of all Pauli strings have $2^n$ values which are $\pm 1$, and the rest zero. This is in contrast to a general state, which will have a wide variation of expectation values of each Pauli string. We can make these stabiliser states by acting on the state $|0\rangle \otimes |0\rangle \otimes \ldots \otimes |0\rangle$ with the particular set of quantum gates listed above.

To the uninitiated, the above definition will be utterly opaque. But it becomes important due to something known as the *Gottesman-Knill theorem* [12]. Roughly speaking, this states

that for any quantum computer containing stabiliser states only, there is a classical computer that is just as efficient! Stabiliser states can include certain maximally entangled states. Thus, something other than entanglement is needed for efficient quantum computing.

The "something else" has been called *magic* in the literature, and from the above comments basically measures "non-stabiliserness" of a quantum state. Different definitions of magic exist in the literature, and we will here use the *Stabiliser Rényi Entropies* of ref. [13]:

$$M_q = \frac{1}{1-q} \log_2(\zeta_q), \quad \zeta_q \equiv \sum_{P \in \mathcal{P}_n} \frac{\langle \psi | P | \psi \rangle^{2q}}{2^n}. \tag{3}$$

In these formulae, $q \geq 2$ is an integer, and $\zeta_q$ represents a weighted sum over the Pauli spectrum values raised to a power. We can think of the set of values $\{M_q\}$ as providing moments of the Pauli spectrum, and this definition turns out to have the following desirable properties: (i) the magic is additive when combining quantum systems; (ii) it vanishes for stabiliser states. In what follows, we will focus on $q = 2$ (i.e. the *Second Stabiliser Rényi Entropy*), given this is already sufficient to quantify non-zero magic. We now have everything we need to investigate magic at the LHC!

## 4   Are top quarks magic?

Given its previous success in probing entanglement, we can look at top pair production as a potential playground for exploring magic. The Standard Model tells us that the most general configuration of top-antitop spins in pair production at the LHC is a *mixed state* (i.e. a superposition of so-called *pure states*). Such states can be described using the density matrix formalism, and the spin density matrix for a top-antitop pair in partonic channel $I$ has the general form

$$\rho^I \sim \tilde{A}^I I_4 + \sum_i \left( \tilde{B}_i^{I+} \sigma_i \otimes I_2 + \tilde{B}_i^{I-} I_2 \otimes \sigma_i \right) + \sum_{i,j} \tilde{C}_{ij} \sigma_i \otimes \sigma_j, \tag{4}$$

where $I_n$ is an $n$-dimensional identity matrix. The quantities $\{\tilde{A}^I, \tilde{B}^{I\pm}, \tilde{C}_{ij}\}$ are called *Fano coefficients*, and are respectively related to the total cross-section, polarisation of the (anti-)top, and spin correlations. Each coefficient depends upon the invariant mass and scattering angle of the top particles, as well as the basis chosen to relate the spin directions $(1, 2, 3)$ to three directions in physical space. A common choice is the *helicity basis* [14], in which one chooses an orthogonal coordinate system aligned with the top quark direction. Defining the normalised Fano coefficients via

$$B_i^{I\pm} = \frac{\tilde{B}^{I\pm}}{\tilde{A}^I}, \quad C_{ij}^I = \frac{\tilde{C}_{ij}^I}{\tilde{A}^I}, \tag{5}$$

the magic of a top quark pair is given by

$$\tilde{M}_2(\rho^I) = -\log_2 \left( \frac{1 + \sum_i [(B_i^{I+})^4 + (B_i^{I-})^4] + \sum_{i,j} (C_{ij}^I)^4}{1 + \sum_i [(B_i^{I+})^2 + (B_i^{I-})^2] + \sum_{i,j} (C_{ij}^I)^2} \right). \tag{6}$$

Figure 1 shows the magic, as calculated in the SM, for both the $q\bar{q}$ and $gg$ initial states. We see that the magic is concentrated away from extreme kinematic limits (e.g. threshold / high energy), which is not surprising: it is known that the top quark final state becomes separable and / or maximally entangled in these regions. These happen to be stabiliser states, and thus the magic vanishes. Note also that the magic can be non-zero where entanglement vanishes, which does not contradict the fact that both entanglement and magic are needed for quantum

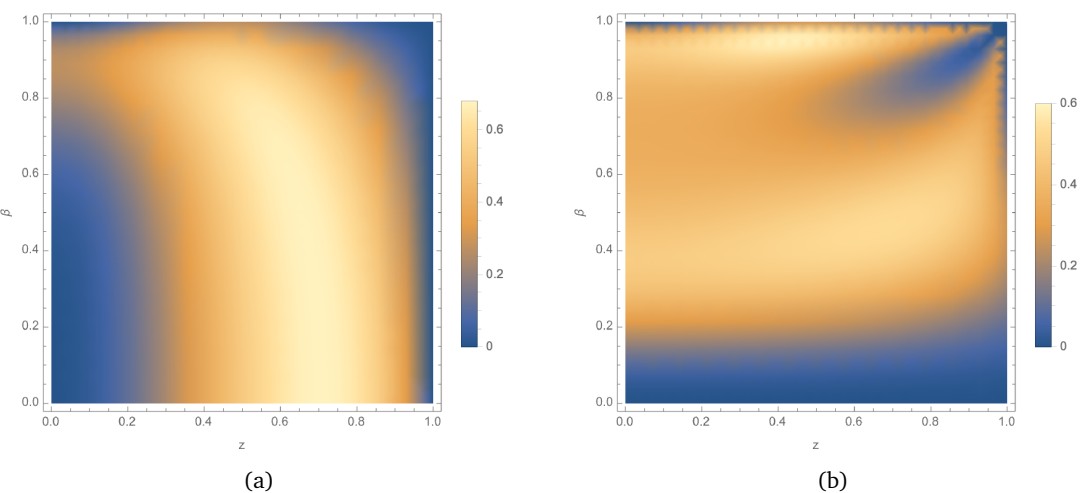

Figure 1: The magic of a mixed top-antitop final state in: (a) the $q\bar{q}$ channel; (b) the $gg$ channel.

computational advantage: the latter is a statement about *algorithms* or *circuits*, which must necessarily contain both entangled and magic states in some intermediate step(s). A given intermediate state, however, does not need to be both entangled and magic.

Results for the magic are also shown for proton-proton initial states in ref. [11]. Combining partonic channels or averaging over angles typically increases magic, due to having a more mixed state, whose Pauli spectrum becomes more complex as a result.

## 5   Conclusion

How to produce and enhance magic in arbitrary quantum systems remains an open research question. We have shown that top quarks provide a system in which magic can be produced, and highly effectively studied using event selection. This may provide insights into how to make other magic systems. Optimistically, one might hope that magic could be used in probing new physics, or in strengthening the already active dialogue between Quantum Computing / collider physics. Further work to investigate these questions is ongoing.

## Acknowledgements

This work was carried out with Martin White, and we are grateful to Rafael Aoude and Hannah Banks for collaboration on related topics.

**Funding information**   This work was supported by the UK Science and Technology Facilities Council (STFC) Consolidated Grant ST/P000754/1 "String theory, gauge theory and duality", and the Australian Research Council grants CE200100008 and DP220100007.

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
