# Peer review of "The magic of top quarks"

_SciPost Physics Proceedings_

## Round 1 · Referee Report · Anonymous (Referee 1) · 2024-12-17

Strengths

These proceedings offer a nice summary of the work carried out in the paper and are good as introduction.

Weaknesses

The claim made in the conclusion regarding that magic could be used to investigate new physics is not sufficiently motivated. In practice, one is measuring polarisations (B's) and spin correlations (C's). It is not shown that the quantity in Eq. 6, obtained from B's and C's, is better than those B's and C's themselves, in order to probe new physics.

Report

I think these proceedings can be published with minimal changes.

Requested changes

  1. Remove the comment about probing new physics.
  2. Is "Martin White" in the acknowledgements the same person as the author "Martin J. White"? I guess so, and it is weird to have an author acknowledging himself.

Recommendation

Publish (easily meets expectations and criteria for this Journal; among top 50%)

---

## Editorial Decision

resubmitted